# QUICKSVIEWER: AN LMM FOR EFFICIENT VIDEO UNDERSTANDING VIA REINFORCED COMPRESSION OF VIDEO CUBES

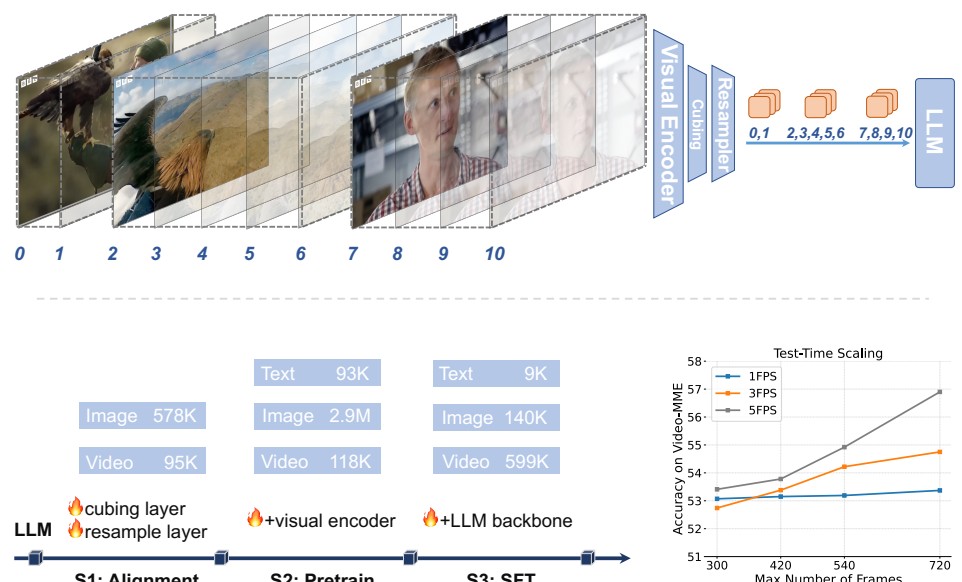

Figure 1: Quicksviewer involves a cubing network that partitions a video into nonuniform cubes, followed by a 3D resampler to gather a fixed number of visual tokens per cube. This efficiency enables **Large Receptive Field** (420 frames) with **High Compression Rate** (64×) during all training stages, and scaling laws on extended frames in inference.

## ABSTRACT

Large Multimodal Models (LMMs) uniformly perceive video frames, creating computational inefficiency for videos with inherently varying temporal information density. This paper present **Quicksviewer**, an LMM with new perceiving paradigm that partitions a video of nonuniform density into varying cubes using Gumbel Softmax, followed by a unified resampling for each cube to achieve efficient video understanding. This simple and intuitive approach dynamically compress video online based on its temporal density, significantly reducing spatiotemporal redundancy (overall 45× compression rate), while enabling efficient training with large receptive field. We train the model from a language backbone through three progressive stages, each incorporating lengthy videos on average of 420s/1fps thanks to the perceiving efficiency. With only 0.8M total video-text samples for training, our model outperforms the direct baseline employing a fixed partitioning strategy by a maximum of 8.72 in accuracy, demonstrating the effectiveness in performance. On Video-MME, Quicksviewer achieves competitive performance compared to models of similar size while utilizing just up to 5% of tokens per frame required by baselines. With this paradigm, scaling up the number of input frames reveals a clear power law of the model capabilities. It is also empirically verified that the segments generated by the cubing network can help for analyzing continuous events in videos.

## 1 INTRODUCTION

Large Multimodal Models (LMMs) (Deepmind, 2024; OpenAI, 2024; Bai et al., 2025) have shown promising progress in video understanding, paying the way for general intelligence in physical world. These models build on Large Language Models (LLMs) and are trained in stages with large-scale image and video data, encoding video frames in the same manner as images before feeding them into the LLM for inference. At the core of these models is the efficient perception of input videos, which is crucial in tackling the persistent contradiction between the **temporal redundancy** of video streams (Buckler et al., 2018; Wenger, 1997) and the **computational efficiency** of LMMs with long context (Fu, 2024).

Extensive studies have been striving to develop LMMs for solving this fundamental issue. Building on devise of frame sampling methods, trailblazing efforts typically involve dedicated token merging strategies (Bai et al., 2025; Wang et al., 2025; Shen et al., 2024; Zohar et al., 2024; Li et al., 2024d; Zhang et al., 2025) and adapted parallel training infrastructures (Zhang et al., 2024b; Chen et al., 2024c; Shen et al., 2025). However, the arbitrary frames sampling and tokens merging introduces inevitable information loss, while marginal compression limits the number of frames in large-scale pre-training.

The velocity of content change in videos is inherently nonuniform, suggesting that the density varies across different temporal cubes. For example in Figure 1, the initial short period features rapidly changing scenes of a researcher attaching a camera to an eagle's back, followed by an extended sequence of stable footage from the camera, and a largely static interview. Inspired by the way that humans adjust their perception speed based on content changes, this paper explores how LMMs can perform video understanding on the nonuniform cubes to achieve dynamic compression, and significantly reduce the spatiotemporal complexity and enhance the efficiency. For practical scenarios where videos typically originate from lengthy offline recordings or online video streams, we thereby aim for the model to **(1) learn from unlabeled data**, perform **(2) online cube partitioning**, and establish a **(3) unified perception paradigm** for images and videos.

We present Quicksviewer, an LMM that perceives nonuniform video cubes for efficient video understanding. Given a video passed from a visual encoder, a small cubing network first partitions it into nonuniform cubes based on the momentum of semantic feature differences between frames, a process that can be conducted online in streaming scenarios. Next, a unified resampling is employed to the cubes to gather a fixed number of tokens for adaptive compression. Finally, these visual tokens, along with absolute timestamps, are fed into the LLM for inference. We integrates the learning of the cubing network into the end-to-end training of the LMM using the Gumbel Softmax (Jang et al., 2016; Herrmann et al., 2020) method with an improved noise annealing mechanism. This reinforced approach not only enables efficient learning on videos without boundary labels but also insures sufficient sampling over the cubes distribution with continuous gradient during training. The nonuniform perception paradigm, which is solely driven by the properties of input video, together with the subsequent resampling enables an efficient video encoding with $45\times$ compression reate, large temporal receptive filed of 420 frames for pre-training, and a consistent representation for both images and videos.

We train our models starting from the LLM backbones through three progressive stages, each incorporating lengthy videos averaging 420s/1fps by benefiting from the efficient perception mechanism. The resulting model, which we coined as Quicksviewer, is an efficient LMM capable of understanding single/multi-images and long videos. We also find that our network is efficient in learning. With only 0.8M video-text samples in total for training, our model achieves competitive performance on Video-MME (Fu et al., 2024a) using just up to 5% of the tokens per frame required by baselines. In addition, to facilitate training on ultra-long videos (*e.g.,* over 1hour), we developed a training infra supporting dynamic changes of sequences lengths based on an existing effort (Chen et al., 2024c), to further facilitate potential explorations in future.

We evaluate Quicksviewer on various video understanding benchmarks, ranging the duration from 16 seconds to 1 hour. Results show that our model outperforms the direct baseline employing a fixed partitioning strategy by a maximum of 8.72 in accuracy, suggesting the utility of the nonuniform perception. We further analyze the cubes partitioning in videos, which demonstrates the emergence of "Visual Lag" phenomenon when the model perceives videos phase-by-phase. We also conduct

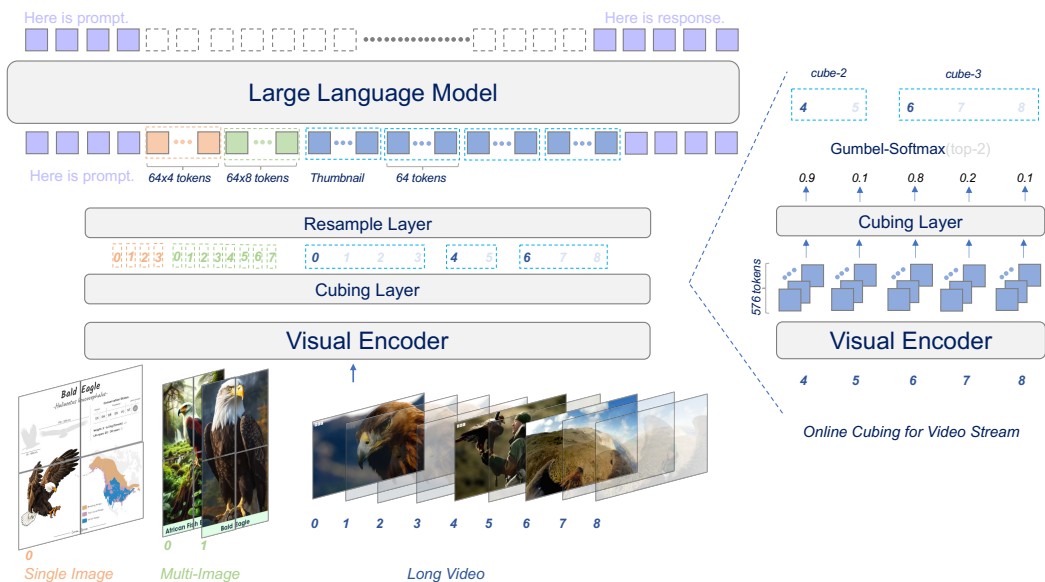

Figure 2: Left: The network architecture of Quicksviewer, that performs unified understanding of videos and images through visual tokens from cascaded modules. Right: The cubing network, that partitions an online video into nonuniform cubes based on Gumbel Softmax.

extensive ablation studies to show the effectiveness of presented components, including the cubing approach, 3D positional encoding, loss penalty, and the annealing strategy.

## 2 APPROACH

The overall architecture of the presented model is shown in Figure 2. We build an efficient LMM that can receive both images and videos as input, where the video is perceived based on nonuniform cubes partitioned through a small cubing network. The model consists of four basic components: (1) a visual encoder $f_v(\cdot)$ that encode image slices or video frames into visual tokens, (2) a cubing network $f_q(\cdot)$ that partitions the video frames into $N_Q$ cubes, (3) a resampler $f_r(\cdot)$ which compress each slice or cube into a fixed number of tokens, and (4) an LLM $f_l(\cdot)$ which accept concatenation of visual tokens and user prompt for response generation. Note that we introduce **FPQ**, the average number of frames per cube, which regulates the percepion granularity and enables adaptive number of cubes $N_Q$ according to video duration.

### 2.1 CUBING USING GUMBEL SOFTMAX

**Visual Encoding** Given an input video to our model, we first uniformly sample $N_F$ frames with a fixed FPS to form $[F_i]_{i=1,..,N_F}$. And then, each frame is firstly encoded using the visual encoder $f_v$ to obtain $N_1$ visual tokens $\mathbf{F}_j = [\mathbf{v}_i]_{i=1,..,N_1}$, where $\mathbf{v}_i \in \mathbb{R}^d$ is the token representation. The large number of visual tokens provides fine-grained semantic representations for each frame, while the concatenation of all tokens (*i.e.,* $N_F \times N_1$) across frames results in an unacceptable sequence length.

**Cubing** It is natural to leverage the semantic difference of features to find keyframes. For compatible with online streaming scenarios and also considering the event-level long-term changes, we utilize the momentum accumulated from the representations of previous frames to track the semantic changes. Given the visual tokens of frames, the cubing network calculate the $i^{th}$ momentum representation as:

$$\Delta_i = \alpha(\mathbf{F}_i - \mathbf{F}_{i-1}) + (1 - \alpha)\Delta_{i-1} \tag{1}$$

where $\Delta_i = [\delta_j]_{j=1,..,N_1} \in \mathbb{R}^{N_1 \times d}$ captures the accumulated semantic changes discounted by factor $\alpha \in [0, 1]$. A 2-layers MLP with LayerNorm (Ba et al., 2016) is then applied to the mean of visual tokens momentums to quantify the significance of the frame:

$$\mathbf{z}_i = \text{MLP}_{\times 2}(\text{LayerNorm}(\frac{1}{N_1}\sum_{j=1}^{N_1}\delta_j)), \qquad \mathbf{z}_i \in \mathbb{R}^2 \tag{2}$$

where the 2-dimensional vector $\mathbf{z} = [w_0, w_1]$ forms a gate, and the sigmoid of their difference directly reflects the keyframe probability $p = \frac{1}{1+\exp{(w_0-w_1)}}$ (Herrmann et al., 2020). We next apply the *top-k* operation on the first dimension of $[\mathbf{z}_i]_{i=2,..,N_F}$ vectors to obtain the indices of $N_Q - 1$ largest values $[l_i]_{i=1,..,N_Q-1}$, which are selected as keyframes. The keyframe $F_{l_i}$, along with its subsequent consecutive non-keyframes, forms the cube $Q_{i+1} = [F_{l_i}, .., F_{l_{i+1}})$. Note that the first sampled frame consistently serves as the keyframe for deriving the first cube $Q_1$.

**Sampling for Training**   During training, we expect the model to perform sufficient sampling exploration while ensuring gradient continuity. We achieve this using Gumbel Softmax with the Straight-Through trick (Gumbel, 1954; Jang et al., 2016):

$$\mathbf{z}_i = \text{softmax}(\mathbf{z}_i - \log(-\log(\epsilon)/\tau)), \qquad \epsilon \sim U(0, 1) \tag{3}$$

where the $\log$ term approximates the sampling process and regulates the degree of exploration.

In experiments, we found that persistent exploration prevents the model from establishing a stable cubing paradigm for subsequent reasoning, leading to sustained loss oscillations. We propose to add an learning rate $\eta$ before the Gumbel noise $\eta \log(-\log(\epsilon)/\tau))$, which is **annealed** from $\eta_0 = 1.0$ to $\eta_T = 0.001$ during training using a cosine scheduler.

## 2.2   RESAMPLING WITH 3D POSITIONAL ENCODING

Based on the partitions from cubing network, a unified 3D resampler is adopted to compress each cube of arbitrary length into a fixed number of $N_2$ tokens.

**Resampling Video Cubes**   We employ the same resampler architecture as (Yao et al., 2024) to compress each cube into a fixed number of dense tokens. We extend the original 2D positional encoding by incorporating a temporal dimension to form 3D position encoding. As a result, each video token is assigned three positional coordinates $(x, y, z)$, representing time, width, and height. We then unfold each cube into tokens sequence along the frame dimension. After adding 3D positional embeddings, an unified resampling is performed to obtain $N_2 = 64$ visual tokens for each cube. For images, we first adopt the AnyRes (Liu et al., 2024a) to divide high resolutional images into slices, and apply the same resampling to each slice to obtain visual tokens and finally concatenate the tokens.

**Resampling for Video Thumbnail**   Using cubes quantized from the cubing network for response tokens generation introduces a fundamental problem: ***How does the NTP training objective optimize the boundary prediction of the cubing network?*** We introduce video thumbnail to resolve this problem meanwhile provide effective global representation. Specifically, we first (1) multiply the 0-1 discretized first dimension of vectors $[\mathbf{z}_i]_{i=1,..,N_F}$ with their corresponding frame representations $[\mathbf{F}_i]_{i=1,..,N_F}$, then (2) average across the frame dimension to obtain $N_1$ visual tokens. A further resampling is performed to get final thumbnail representation containing $N_2$ tokens. This simple approach allows gradients to be directly propagated back to the cubing boundaries. The final representation of a video is a concatenation of the representations of the thumbnail and cubes.

## 2.3   LLM INFERENCE WITH AUXILIARY LOSS

Following resampling of nonuniform video cubes, the tokens of each cube span varying temporal windows. We prepend each cube with an absolute timestamp as a float number in 0.01-second units,

enabling explicit temporal awareness. We also enclose the video, thumbnail, and image tokens with their corresponding special tokens to enable explicit content differentiation.

During training, we observed that excessively large values of $\mathbf{z}$ cause overly high gradients, impairing convergence. To address this, we introduce an auxiliary $L_2$ norm loss with $\beta = 0.001$ penalty weight on them to constrain its values within a reasonable range.

## 3 TRAINING PROCESS

We train our models with three progressive stages starting from LLM backbones, each stage incorporating lengthy videos on average of 420s by benefiting from the efficient perception approach.

**Stage-1: Multimodal Alignment** We utilize both interleaved and captioning image-text corpuses, and video-text captioning corpus to train our models, establishing fundamental alignment between visual encoder and LLM backbones with in-context learning capabilities. We sample a subset of 20K sequences from OBELICS (Laurençon et al., 2023), with each containing more than two interleaved pairs. We utilize LCS (Li et al., 2024a), a re-captioned dataset consisting of 558K detailed descriptions from the CC3M (Sharma et al., 2018). The video-text training data incorporates a sampled subset of 87K captioning pairs from FineVideo (Farré et al., 2024) and 8K captaining pairs from ANetCaptions (Krishna et al., 2017). We train parameters of the cubing network and resampler while keeping all other parameters frozen to establish a stable projection. The models are trained for 1 epoch with a $lr$ that warms up to $1e^{-4}$ over the first 2% of steps, then gradually decays to 0.

**Stage-2: Pre-training** We employ large-scale pretraining data, primarily consisting of image-text multi-task data, to pre-train models establishing general multimodal capabilities across broad visual scenarios. We utilize a subset of 2.99M samples from LLaVA-OneVision-SingleImage (Li et al., 2024a) as training corpus, which incorporates 2.9M image-text pairs and 93K textual instruction-tuning samples from Evo-Instruct (Chen et al., 2024a). For video-text corpus, we utilize a sampled subset of 75K video QAs from FineVideo (Farré et al., 2024) and 38K captioning pairs from ShareGPT4Video (Chen et al., 2024b). To mitigate catastrophic forgetting, we retain 5% of the previous image and video data in our training corpus. Alongside the cubing network and resampler, we also unfreeze the visual encoder to improve the visual representation. We train models for 1 epoch with a $1e^{-5}$ initial learning rate, with the same warmup and decay schedule as stage-1.

**Stage-3: Supervised Fine-tuning** We primarily leverage extensive video-text paired corpus to train our models in this stage, enabling robust video understanding capabilities. We primarily utilize a subset of 476K video-text samples sourced from VideoChat2-IT (Li et al., 2024c), and a subset of 79K samples from ShareGPTVideo (Zhang et al., 2024c) as the video corpus. To enhance adaptation to long video scenarios, we further integrate 5K samples from MovieChat (Song et al., 2024) and 39K samples derived by (Chen et al., 2024c) from the Shot2Story dataset (Han et al., 2023). The image-text corpus incorporates a sampled subset of 100K mult-image, multi-task understanding samples from LLaVA-OneVision-MultiImages (Li et al., 2024a). We also preserve a subset of training data from the previous stage, consisting of 40K text-image pairs and 9K textual instruction-tuning

| | | Stage-1 | Stage-2 | Stage-3 |
|---|---|---|---|---|
| *Vision* | **Resolution** | $384 \times 384$ | $384 \times 384$ | $384 \times 384$ |
| | FPS,#Frames | 1, Max 420 | 1, Max 420 | 1, Max 420 |
| *Data* | **Image-Text** | LCS, OBELICS | LLaVAOV-SingleImage | LLaVAOV-MultiImages |
| | #samples | 558K, 20K | 2.99M | 100K |
| | **Video-Text** | FineVideo, ANetCaptions | FineVideo, ShareGPT4Video, ANetCaptions | Sec. 3. 3 |
| | #samples | 87K, 8K | 118K | 599K |
| *Model* | **Trainable** | Cubing, Resampler | Cubing, Resampler, ViT | Full Model |
| | #Parameters | 75M | 500M | 8B |
| *Training* | **Anneal:** $\eta_0, \eta_T$, ratio | 1.0, 0.01, 0.8 | 1.0, 0.01, 0.6 | 1.0, 0.01, 0.6 |
| | **LR:** $\theta_c, \theta_r, \theta_v, \theta_l$ | $1e^{-4}, 1e^{-4}$,-,- | $2e^{-5}, 2e^{-5}, 2e^{-5}$,- | $1e^{-5}$ |
| | **Epoch** | 1 | 1 | 1 |

Table 1: Detailed configuration for each training stage.

samples. We train all parameters for 1 epoch with a learning rate that warms up to $1e^{-5}$ over the 0.02 epoch, followed by gradual decay to 0 for the remaining duration.

During training, all videos sampled at 1FPS to extract full frames. For videos exceeding 420s, we uniformly extract 420 frames to maintain computational tractability. Images are processed using AnyRes with a resolution of $384 \times 384$. For all stages, the Gumbel noise learning rate $\eta$ (initialized at 1.0) undergoes cosine annealing to 0.01 within: 0.8 epoch (Stage 1) or 0.6 epochs (Stages 2-3).

# 4 EXPERIMENT

| Models | Size | #Tokens | #Train | MMBench-Video | MVBench | MLVU | Video-MME |
|---|---|---|---|---|---|---|---|
| Duration | | /Frame | V-T | 3 min | 16 sec | 3∼120 min | 1∼60 min |
| *Proprietary Models* | | | | | | | |
| GPT4-V (OpenAI, 2023) | - | - | - | 1.53 | 43.7 | - | 60.7 |
| GPT4-o (OpenAI, 2024) | - | - | - | 1.63 | 64.6 | 66.2 | 77.2 |
| *Open-Source Video LMMs* | | | | | | | |
| LLaMA-VID (Li et al., 2024e) | 7B | 2 | 0.4M | 1.08 | 41.5 | 33.2 | - |
| LongLLaVA (Wang et al., 2024c) | 9B | 144 | 0.5M | - | 49.1 | - | 43.7 |
| Chat-UniVi (Jin et al., 2024) | 7B | 112 | 100K | 1.06 | 42.9 | - | 45.9 |
| ShareGPT4Video (Chen et al., 2024b) | 8B | 144 | 4.8M | 1.05 | 51.2 | 46.4 | 43.6 |
| LLaVA-NeXT-Video (Zhang et al., 2024d) | 7B | 144 | 100K | 1.14 | 33.7 | - | 46.5 |
| VideoLLaMA2 (Cheng et al., 2024) | 7B | 32 | 10.7M | 1.08 | 54.6 | 48.5 | 46.6 |
| LongVA Zhang et al. (2024b) | 7B | 144 | - | - | - | 56.3 | 54.3 |
| VideoChat2 (Li et al., 2024c) | 7B | 64 | 2.8M | 1.22 | **60.4** | 47.9 | 54.6 |
| mPLUG-Owl3 (Ye et al., 2024) | 8B | 729 | 134K | **1.35** | 54.5 | - | 53.5 |
| Fixed-LLama3.1 | 8B | 12.8 | 0.8M | 0.71 | 45.2 | 50.2 | 45.0 |
| Quicksviewer-LLama3.1 | 8B | 12.8 | 0.8M | 0.87 | 53.9 | 58.6 | 47.6 |
| Quicksviewer | 8B | 12.8 | 0.8M | 1.24 | 55.6 | **61.5** | **56.9** |

Table 2: Video benchmarking results between Quicksviewr and baselines under comparable total sequence length. Quicksviewer achieves multiple SOTA performance while using fewer tokens per frame (up to 5% of baseline) and substantially less video-text training samples.

## 4.1 IMPLEMENTATION DETAILS

We use SigLIP (Zhai et al., 2023) (soo400m-path14-384) as our visual encoder inconsistent with previous works. We adopt Qwen2.5 (Yang et al., 2024) as the language backbone for our standard implementation (i.e., Quicksviewer), while utilizing Llama3.1 (Touvron et al., 2023a) as the alternative LLM for another version (i.e., Quicksviewer-Llama3.1) for comprehensive exploration. We use AdamW (Loshchilov & Hutter, 2017) optimizer with a cosine scheduler for all training stages. The number of tokens generated from visual encoder and resampler are $N_1 = 576$, and $N_2 = 64$, respectively. The discounting factor of momentum is set to $\alpha = 0.9$. The penalty weight to the auxiliary loss is set to $\beta = 0.001$. We use FPQ=5 for all models. Our models is trained on 48 NVIDIA A100 GPUs.

## 4.2 EXPERIMENTS ON VIDEO UNDERSTANDING

We train a direct baseline, Fixed-Llama3.1, which utilizes uniform temporal partitioning with the same FPQ of input videos. For an unbiased comparison, we evaluate with baselines configured with comparable total sequence lengths, maintaining equivalent computational budgets.

**Benchmarks and Metrics** We evaluate the our models on widely used video understanding benchmarks Video-MME (Fu et al., 2024a), MVBench (Li et al., 2024c), and MLVU (Zhou et al., 2024) to investigate the effectiveness. VideoMME is a general video understanding benchmark that collect videos (1min∼1hour) from Youtube with manual annotations. MVBench covers 20 challenging tasks ranging from perception to cognition. MLVU (3mins∼2hours) refers to an long video understanding benchmark for long-term inference.

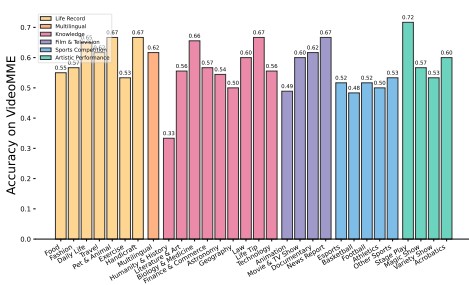 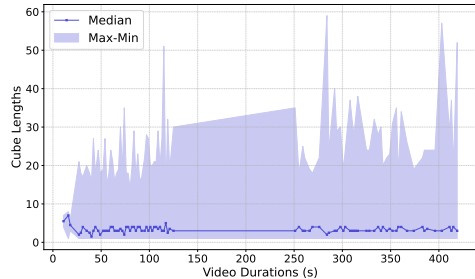

Figure 3: (a) Left: Performance of Quicksviewer on particular domains and categories of Video-MME. (b) Right: Distribution of cube lengths across Video-MME videos.

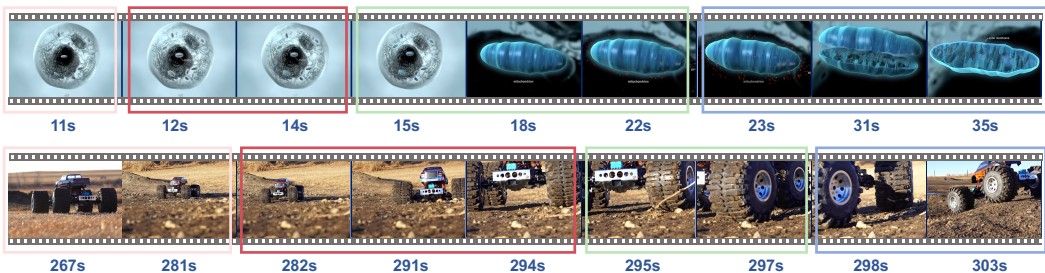

Figure 4: The **"Visual Lag"** phenomenon occurring during the model's cube-based segmental comprehension, where current cubes incorporate terminal frames from preceding event scenes to enable retrospective understanding.

**Quantitative Results**    We adopt empirically optimized configuration: 5 FPS with a maximum of 720 frames for all benchmarks. As the main result shown in Table 2, our standard model achieves competitive performance on Video-MME and MLVU, and the competitive performance on MVbench with significantly fewer tokens and training volumes. Specifically, our model achieves competitive performance on Video-MME, albeit with slightly inferior results on long videos. This demonstrates that the encoding paradigm harness the scaling benefits by high frame rate. In comparison with the direct baseline with fixed cubing strategy, our model obtain large improvements, suggesting the effectiveness of the cuing strategy. Our approach achieves competitive performance on MLVU while demonstrating competitive performance on MVBench, despite utilizing substantially less training data (only 28% of VideoChat2's and 7.5% of VideoLLaMA2's requirements). This evidences our network's exceptional learning efficiency.

**Analysis**    We further analyze the model performance across distinct domain categories in Video-MME, systematically examine both capability advantages and limitations. As illustrated in Figure 3 (a), bars sharing identical colors belong to the same domain. Primarily, we observe consistent model performance across all domains, with mean scores of 0.61, 0.62, 0.55, 0.60, 0.51, and 0.61 respectively, suggesting limited domain-specific variation in question difficulty. Secondly, the model demonstrates suboptimal performance (below 50%) in three categories: Humanity & History, Animation, and Basketball. This may indicate persistent challenges in fine-grained character recognition that require further improvement.

We further analyze the distribution of cube lengths on Video-MME, with results shown in Figure 3 (b). Based on the predefined FPQ, we found the median cube length approximates 5 frames. Notably, the model demonstrates a tendency to partition diverse length of cubes for longer videos, which aligns with the variable viewing speeds in human perception of lengthy videos.

## 4.3    ANALYSIS OF THE CUBES PARTITIONING

To investigate how the trained model partitions cubes for understanding, we analyzed two representative video cases by examining cubes relative to content transitions. As shown in Figure 3, each box

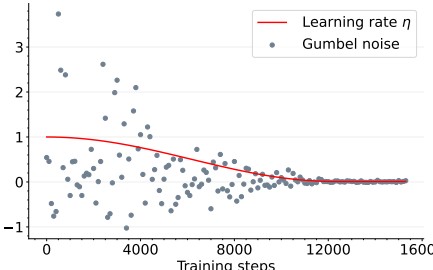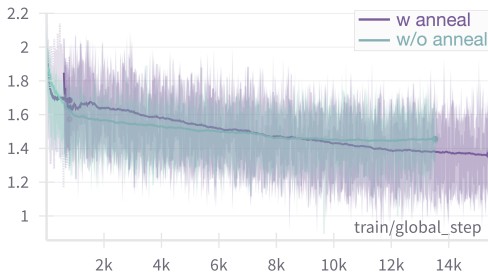

Figure 5: (a) Left: Gumbel noise progressively anneals to 0.001 following the decaying learning rate with cosine scheduler. (b) Right: Compared to non-annealed training (cyan curve), adding Gumbel noise annealing (purple curve) yields more stable and superior loss convergence.

refers a cube spanning from its start to end timestamps (with similar intermediate frames omitted). We reveal a **"Visual Lag"** phenomenon during cube-based video perception of the model: terminal frames from preceding event scenes are incorporated into cubes containing subsequent event scenes. For example in the first video, the initial frames of cubes 2-4 respectively contain content from three event scenes: 1) cellular details, 2) mitochondrial positioning, and 3) ATP synthesis exhibiting in cubes 1-3 respectively. We posit this mechanism enables the model to retain partial memory of preceding scenes to facilitate current scene understanding.

## 4.4 ANALYSIS OF THE ANNEALING STRATEGY

Traditional Gumbel-Softmax training controls sampling randomness exclusively through temperature adjustment, making it unsuitable for training the cubing network as a component of an LMM. To resolve this issue, we propose annealing the Gumbel noise, which substantially improves both training stability and effectiveness. To further evaluate the performance of the proposed annealing mechanism, we examine the evolution of Gumbel noise values throughout a training epoch in with the annealed learning rate. For clear visualization, we uniformly sample training steps at 100-step intervals, as illustrated in Figure 5 (a). From the figure, we observe that in the early training stages, larger Gumbel noise effectively facilitates exploration for the cubing network. As training progresses, the Gumbel noise gradually converges to the predefined value of 0.01. This allows the model to leverage its learned segmentation mechanism for video understanding in later stages, stabilizing the training process and achieving optimal performance.

Figure 5 (b) compares the loss trajectories of models trained with and without the annealing mechanism. Initialized from the same checkpoint from Stage-2, we train parallel models using both approaches and monitor loss variations throughout one epoch to assess learning efficiency and stability. Our analysis reveals that the model benefits from the progressive annealing of Gumbel noise in the later stages. During this phase, the model effectively utilizes its learned cubing mechanism to accelerate loss minimization, achieving superior convergence efficiency, demonstrating the effectiveness and training stability.

## 4.5 ABLATION STUDIES

We conduct comprehensive ablation studies to evaluate the efficacy of the components leveraged in Quicksviewer. To establish a simple baseline, we first train a Llama3.1 (Touvron et al., 2023a) model through Stages 1-2 using only image-text data introduce in Sec. 3, deliberately excluding video inputs. This image-only pretrained checkpoint then serves as the initialization point for systematically investigating various Stage-3 configurations with video-text data.

**Cubing network with ViT** To accelerate cube processing, we investigate the feasibility of using only the initial n layers of ViT for the cubing network. Our ablation study employs the first 2 ViT layers for cube feature generation while maintaining all other model components unchanged. As demonstrated in Table 7, this configuration results in significant performance degradation, indicating that shallow visual features are insufficient for effective cubes partitioning.

| Ablation Components | | | | | Video-MME | |
|---|---|---|---|---|---|---|
| | | | | | Overall | Long |
| PE | Cubing Network | Penalty | Annealing | Trainable | 1∼60 min | 30∼60 min |
| 2D | ViT first 2-layers | 0.1 | - | $\theta_c, \theta_r, \theta_v$ | 33.92 | 35.56 |
| 2D | ViT full | 0.1 | - | $\theta_c, \theta_r, \theta_v$ | 41.22 | 38.67 |
| 3D | ViT full | 0.1 | - | $\theta_c, \theta_r, \theta_v$ | 44.37 | 40.67 |
| 3D | ViT full | 0.001 | - | $\theta_c, \theta_r, \theta_v$ | 44.66 | 40.44 |
| 3D | ViT full | 0.001 | annealing | $\theta_c, \theta_r, \theta_v$ | 45.44 | 43.44 |
| 3D | ViT full | 0.001 | annealing | All | **45.96** | **38.44** |

Table 3: Ablation results of Stage-3 training initialized from a checkpoint pretrained only with image data (Stage 1-2). The optimal configuration: 3D positional encoding, Gumbel noise annealing with 0.001 penalty weight, and full trainable parameters, demonstrating superior performance.

**3D positional encoding**   We systematically evaluate the impact of 3D positional encoding compared to the original 2D formulation. Implementing this modification while keeping all other parameters fixed in Stage-3 training, our experiments demonstrate a consistent accuracy improvement of +3.15% (Table 7), confirming the benefits of spatiotemporal position awareness for video understanding.

**Penalty weight to the auxiliary loss**   The penalty weight applied to the $L_2$-norm of $\mathbf{z}$ values regulates the scale of logistic outputs, consequently influencing partition behavior. Through systematic ablation while maintaining other parameters fixed, we identify $\beta = 0.001$ as the optimal weighting that simultaneously: (1) maintains $\mathbf{z}$ values in an effective operational range, and (2) achieves superior model performance (Table 7).

**The annealing strategy**   Having established the optimal penalty weight, we proceed to evaluate the efficacy of our proposed Gumbel noise annealing strategy. This approach systematically reduces exploration randomness during training, transitioning from aggressive parameter space exploration to fine-tuned optimization. Comparative results in Table 7 demonstrate consistent performance improvements over the fixed-noise baseline, validating the benefits of noise scheduling.

**The tuning parameters**   We examine the impact of trainable parameters on video understanding by comparing two training regimes: (1) our baseline approach that only fine-tune the ViT and resampler parameters during Stage-3, versus (2) a full-parameter optimization strategy that additionally fine-tunes the LLM backbone. As evidenced in Table 7, comprehensive parameter training yields superior benchmark performance, suggesting that joint visual-linguistic optimization enhances multimodal alignment for video understanding tasks.

## 5   CONCLUSION

In this paper, we introduced Quicksviewer, an LMM designed for efficient video understanding through a nonuniform perception paradigm. By dynamically partitioning videos into nonuniform cubes and applying adaptive resampling, our approach achieves a 45× compression rate while maintaining a consistent representation for both images and videos. We demonstrated that integrating the cubing network into end-to-end training via Gumbel Softmax with an improved noise annealing mechanism, enables efficient learning without boundary labels. Furthermore, our model, trained on just 0.8M videos, achieves competitive performance on VideoMME with significantly fewer tokens per frame than the baseline methods. To support training on ultra-long videos, we also developed an infra that allows dynamic sequence lengths. These contributions pave the way for efficient and scalable LMMs, facilitating future research in long video understanding.

Our approach demonstrates strong potential for future applications in video segmentation and long-sequence processing. In video tasks, cubing-based networks enable fine-grained segmentation of videos into multiple events, supporting effective video preprocessing. Moreover, our method is highly generalizable and can be adapted for various sequence segmentation and compression scenarios, providing a foundation for efficient long-sequence analysis.

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

# Appendix

## A  RELATED WORKS

**LMMs for General Vision Understanding**  Building upon pretrained large language models (Brown et al., 2020; Touvron et al., 2023a;b) and vision foundation models (Radford et al., 2021; Zhai et al., 2023) with exceptional capabilities, Large Multimodal Models (LMMs) (Alayrac et al., 2022; Li et al., 2023a; Liu et al., 2023; Bai et al., 2023; Wang et al., 2024b; Yao et al., 2024) were initially proposed to learn mappings from visual inputs to linguistic representations through training on vast amounts of image-text paired data. These models have demonstrated remarkable visual-language understanding abilities. Subsequent studies has witnessed significant advancements in this field, with notable contributions including superior visual grounding Qi et al. (2024); Wang et al. (2024a), for web content recognition and visual agents (Hong et al., 2024b), enhanced multi-modal reasoning capabilities (Chen et al., 2024e;d; Fu et al., 2024b; 2025; Shi et al., 2024; Zhang et al., 2024a), and the efforts contributed substantially to the open-source community (Lin et al., 2024; Liu et al., 2024b; Li et al., 2024a;b). These developments collectively represent substantial progress in expanding the applicability and performance of multimodal systems across diverse scenarios. Recent research has witnessed a surge of work developing LMMs with video understanding capabilities by incorporating video frames as training data (Chen et al., 2024d; Hong et al., 2024a; Ye et al., 2024; Yang et al., 2024; Wang et al., 2022; 2024d; Guo et al., 2025; Li et al., 2025; 2023b; 2024c).

**LMMs for Efficient Video Understanding**  To handle longer video inputs, numerous efficient LMMs have been proposed. Early optimizations drew inspiration from long-context techniques in language models, including the adoption of sequence parallelism frameworks (Zhang et al., 2024b; Xue et al., 2024; Shen et al., 2025). Subsequently, a series of token merging techniques were developed to reduce spatiotemporal redundancy by consolidating the expanded token sequences resulting from longer video input (Li et al., 2024e; Maaz et al., 2023; Xu et al., 2024a; Shen et al., 2024; Sun et al., 2024; Shu et al., 2024; Ren et al., 2023). Additionally, significant research efforts have focused on video frame sampling strategies to minimize inter-frame redundancy (Ataallah et al., 2024; Zhang et al., 2024e; Xu et al., 2024b; Zohar et al., 2024; Chen et al., 2024b; Chang et al., 2024). More recently, novel architectures with enhanced long-sequence memory capabilities have emerged to facilitate the understanding of extended video sequences (Wang et al., 2024c; Islam et al., 2025).

## B  LIMITATIONS

This paper explores efficient multimodal large models for video understanding. Specifically, we train a video comprehension model from scratch using moderately-sized datasets based on a language model architecture. Benefiting from a dynamic compression-based efficient encoding scheme, the model processes video frames with high receptive fields during training and demonstrates competitive performance across multiple benchmarks during inference. While comprehensive ablation studies validate the model's effectiveness, two key limitations persist: (1) Training data quality - we observe that most pretraining video captions are summary-type texts, which hinders the model's ability to learn detailed video content understanding; (2) Temporal alignment of visual positional encoding - we identify misalignment between the model's visual positional encoding and the actual temporal sequence during training, leading to inaccuracies in temporal reasoning during inference. There is a potential solution to solve the temporal misalignment problem which we have initially validated the effectiveness. The solution is that 1) we first calculate the start and end time boundaries for each cube (e.g., 0–6 seconds for the first cube), and then 2) uniformly sample PE indices within the interval (e.g., np.linspace(0, 6, 64)), and finally 3) these interpolated or extrapolated indices are finally used to compute the PE embeddings based on M-ROPE. This approach enables more precise temporal alignment. These limitations will be addressed in our future work.

## C TRAINING DATA DETAILS

| Modality | Task | # Samples | Dataset |
|---|---|---|---|
| Image-Text | Interleaved Pairs | 20K | OBELICS |
| | Single-Image Captioning | 558K | LCS |
| Video-Text | Captioning | 95K | FineVideoCaptions, ANetCaptions |

Table 4: Training data statistics for the alignment stage.

| Modality | Task | # Samples | Dataset |
|---|---|---|---|
| Text | Instruction | 93K | Evo-Instruct |
| Image-Text | Interleaved Pairs | 20K | OBELICS |
| | Single-Image Captioning | 50K | LCS |
| | Single-Image Tasks | 2.8M | LLaVAOneVision |
| Video-Text | Captioning | 5K | FineVideoCaptions, AnetCaptions |
| | VQA | 75K | FineVideoQAs |
| | Dense Captioning | 38K | ShareGPT4Video |

Table 5: Training data statistics for the pre-training stage.

| Modality | Task | # Samples | Dataset |
|---|---|---|---|
| Text | Instruction | 9K | Evo-Instruct |
| Image-Text | Single-Image Tasks | 40K | LLaVA-OneVision-SingleImage |
| | Multi-Images Tasks | 100K | LLaVA-OneVision-MultiImages |
| Video-Text | Captioning | 52K | TextVR, MovieChat, YouCook2 |
| | Dense Captioning | 4K | ShareGPT4Video |
| | Classification | 1K | Kinetics-710 |
| | VQA | 354K | NExT-QA, CLEVRER, EgoQA TGIF, ShareGPTVideo, FineVideoQAs |
| | Instruction | 188K | VideoChatGPT, VideoChat, LongVILA |

Table 6: Training data statistics for the supervised fine-tuning stage.

# D  ADDITIONAL EVALUATIONS

| Model | NExT-QA | ActivityNet-QA | Video-ChatGPT | | | | |
|---|---|---|---|---|---|---|---|
| | acc | acc | Correctness | Detail | Context | Temporal | Consistency |
| LLaMA-VID (7B) | - | 47.4/3.3 | 2.96 | 3.00 | 3.53 | 2.46 | 2.51 |
| Chat-UniVi (7B) | - | 46.1/3.3 | 2.89 | 2.91 | 3.46 | 2.89 | 2.81 |
| Video-LLaVA (7B) | - | 45.3/3.3 | 2.87 | 2.94 | 3.44 | 2.45 | 2.51 |
| VideoChat2 (7B) | 68.6 | 49.1/3.3 | 3.16 | 3.08 | 3.69 | 2.56 | 3.14 |
| VideoLLaMA2 (7B) | 75.6 | 50.2/3.3 | 3.30 | 33.18 | 3.78 | 2.66 | 3.12 |
| LLaVA-NeXT-Video (7B) | 78.2 | 53.5/3.2 | 3.39 | 3.29 | 3.92 | 2.60 | 3.12 |
| Quicksviewer (8B) | 77.5 | 47.6/2.7 | 3.10 | 3.11 | 3.09 | 2.48 | 3.04 |

Table 7: Evaluation results on more benchmarks.

# E  QUALITATIVE ANALYSIS

Our unified perception paradigm demonstrates efficient visual understanding capabilities, successfully processing: 1) lengthy videos, 2) high-resolution single images, and 3) multi-image contextual reasoning tasks. Through representative qualitative analysis, we validate the model's performance across these diverse inpus. Specifically, we evaluate the video understanding capabilities of our model through documentary recording and sports competitions, while the image understanding proficiency across diverse domains including outdoor road scenes, physics/biological/historical knowledge systems, and multi-image geographical analysis.

In the documentary depicting a penguin chick's perilous encounter and subsequent escape, our model demonstrates comprehensive video understanding by: (1) identifying the nature of the unexpected attack, (2) precisely locating its temporal occurrence, and (3) summarizing the eventual outcome - showcasing its advanced capabilities in long-form video understanding, including temporal action recognition, event narrative abstraction, and exact timestamp localization.

In a lacrosse match video documenting a scoring play, our model precisely identifies the initiating player when queried about the offensive sequence, subsequently describing the play development and correctly specifying both the scoring player's identity and jersey number. This demonstrates the model's dual capability of (1) recognizing individual athletes in sports footage and (2) logically summarizing dynamic game situations.

We further validate our model's image understanding capabilities across extensive scenarios. As illustrated in Figure 6, these includes: 1) traffic sign recognition in driving environments, 2) physics problem solving, 3) biological image interpretation and association, 4) historical scene identification, and 5) multi-image geographical reasoning. These examples demonstrate that while exhibiting strong video understanding, our model maintains robust image understanding capabilities. Benefiting from interleaved image-text training data, the model additionally acquires in-context learning capabilities for complex reasoning tasks.

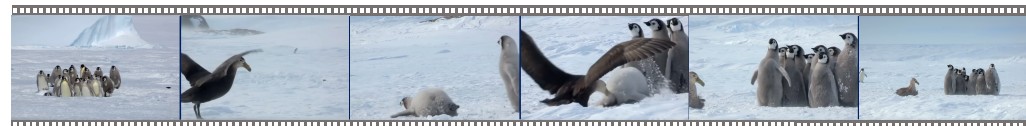

**User**: What is the main incident that happened in this video?

**Quicksviewer:** The main incident that happened in the video is a group of penguins standing on the snow, while a bird approaches them. The bird tries to attack one of the penguins, but it fails.

**User**: During what time period (HH:MM:SS - HH:MM:SS) did the attack occur? Were any penguins injured?

**Quicksviewer:** The attack occurred between 0:47 and 0:53, and no penguins were injured during the incident.

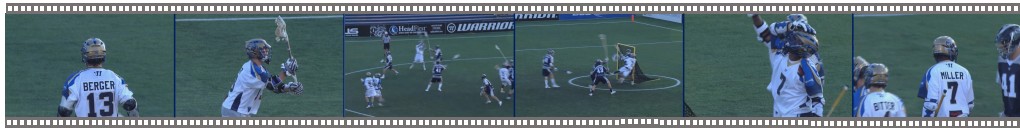

**User:** Which White team player started the last offensive sequence?

**Quicksviewer:** 13, Barger.

**User:** What happened next?

**Quicksviewer:** He passed the ball to his teammate, who then passed it to another teammate. White 7 Miller shot and scored a goal. The teammates celebrated by hugging and cheering.

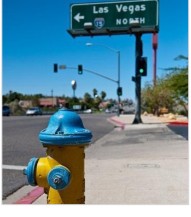

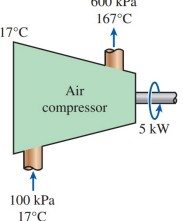

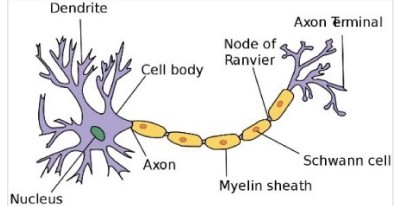

**User:** Where can I go by turning left?

**Quicksviewer:** Las Vegas.

**User:** Describe the process shown in the diagram including the specific values.

**Quicksviewer:** The thermodynamic process involves an air compressor of 5kW. The input values are 100 kPa and 17°C, while the output values are 167 kPa and 176°C.

**User**: Identify the biological structure and specify its computational counterpart in CS disciplines.

**Quicksviewer:** The image depicts a neuron, a fundamental structure in nervous tissue. The corresponding concept in computer science would be the neuron in artificial neural networks.

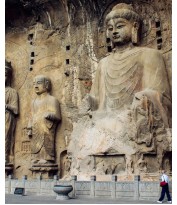

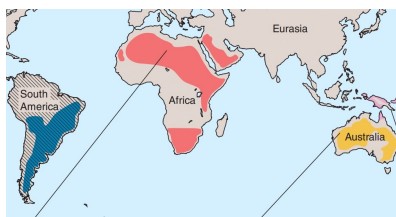

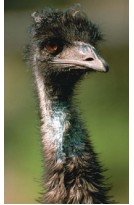

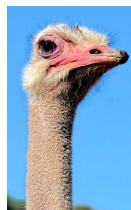

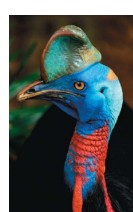

**User:** Where is it located?

**Quicksviewer:** This is the Longmen Grottoes, located in Luoyang, Henan, China.

**User:** What are the species of the birds, and which one inhabits the red area marked in the first map?

**Quicksviewer:** The first bird is a South American ostrich, the second is an African ostrich, and the third is a cassowary. The African ostrich inhabits the red area marked in the first map.

Figure 6: Qualitative analysis showns that Quicksviewer effectively understands lengthy documentary and sports videos, as well as informative single and multiple images.

