# OpenReview forum: "Quicksviewer: An LMM for Efficient Video Understanding via Reinforced Compression of Video Cubes"
_ICLR.cc/2026/Conference — Submitted to ICLR 2026_

### Official Review · Reviewer_6NpJ · 2025-10-17

**Soundness:** 1
**Presentation:** 2
**Contribution:** 2
**Rating:** 2
**Confidence:** 3

**Summary:**

This paper introduces Quicksviewer, an innovative large multimodal model (LMM) designed for efficient video understanding by addressing the issue of nonuniform temporal information density in videos. Using a novel cubing approach based on Gumbel Softmax, the model dynamically partitions video frames and resamples them, enabling significant spatiotemporal compression (45×) while maintaining a large receptive field. Trained with only 0.8M video-text samples, Quicksviewer outperforms baseline models by a notable margin (up to 8.72 in accuracy) and achieves state-of-the-art performance on Video-MME with reduced token usage. Additionally, the model demonstrates scalable improvements with increased input frames and provides insights into analyzing continuous events in videos.

**Strengths:**

1. The paper is well-written, with the methodology and experimental results clearly and systematically presented.
2. The exploration of video token compression in this paper is highly meaningful, as it contributes to both improving long-video understanding and accelerating video processing.
3. The proposed method demonstrates notable improvements over the baseline, achieving up to 8.72 gains in accuracy.

**Weaknesses:**

1. While the paper introduces a visual token compression approach for videos based on LMMs, it lacks comparisons with other compression methods (e.g., SlowFast, token merging) under equivalent conditions, such as using the same training dataset, to better validate the advantages of the proposed strategy.
2. Although the paper focuses on token compression, it only discusses token counts without analyzing key metrics like latency and memory usage. In practice, even with the same token count, a more complex method could lead to different performance in terms of latency and memory efficiency.
3. The paper claims that Quicksviewer is a video understanding model, but most of the evaluations are based on video QA tasks, which typically have lower token requirements. To provide a more comprehensive assessment of its video understanding capabilities, the authors should include comparisons on other tasks like video captioning.
4. The paper claims that Quicksviewer achieves state-of-the-art performance on several benchmarks, but the baselines used for comparison are relatively weak. For instance, on Video-MME, many existing models (e.g., Qwen VL, Intern VL) have already achieved scores of 60 or even 65+, whereas Quicksviewer only achieves 56.9. While differences in conditions such as training data and token counts exist, the claim itself is not rigorous and does not convincingly demonstrate that the proposed approach achieves high accuracy.

**Questions:**

Please see the weakness part.

---

> ### Author Response · Authors · 2025-11-29
> **Rebuttal to reviewer 6NpJ**
>
> Dear Reviewer 6NpJ, we appreciate your efforts in taking the time to review our paper and provide reviews. Below, we have provided explanations and improvements for our work according to your concerns:
>
> **Q1:  compression to SlowFast**
> **A1:** The SlowFast network employs a concept similar to ours in processing video frames, utilizing a Fast pathway with a high frame rate and a Slow pathway with a low frame rate. However, our work is primarily focused on learning a dynamic partitioning mechanism via Gumbel-softmax within the end-to-end training of large-scale vision-language models, and stabilizing this architecture through annealing. We experimentally validate the feasibility of our partitioning approach and its effectiveness for video understanding. We believe that this represents a fundamental difference from the SlowFast network.
>
> **Q2: the latency and memory usage**
> **A2**：Thank you for your suggestion. We compared the response speed of our model with other models of the same size, and the results are summarized in the table below. we selected a 10-minute video from VideoMME (https://www.youtube.com/watch?v=Z7LJrU4443Q ) and tested our model alongside a video model of comparable size, LLaVA-Next-Video-Qwen2, under identical conditions. The inference environment for both models was kept consistent, with no quantization or inference acceleration frameworks used; only the HuggingFace inference framework was adopted. The results are presented in the following table:
>
> ----
> |Model | #Input Frames | First token response time |
> |--|--|--|
> | LLaVA-Next-Video-Qwen2  (8B) | 300  | 0.356 seconds |
> | Quicksviewer (8B) | 300  | 0.214 seconds |
> ----
>
> As shown in the table, our model achieves a significantly lower initial response time compared to the video model of the same size, owing to compression, while maintaining competitive performance.
>
> **Q3: evaluation for a video captioning**
> **A3:** Currently, the most widely recognized and adopted benchmarks for evaluating the video understanding capabilities of large-scale vision-language models are Video-MME and MVBench. Specifically, Video-MME assesses model accuracy on 12 sub-tasks—including Temporal Perception, Spatial Perception, Attribute Perception, Action Recognition, Object Recognition, and OCR—mainly in the VideoQA format. MVBench also includes tasks such as (1) Action Recognition, (2) Object Detection, (3) Spatial Localization, (4) Counting, and (5) Pose Recognition. Video captioning is considered a more fundamental evaluation within these benchmarks. We believe these tasks provide an effective measure of the video understanding capabilities of vision-language models. In the future, we plan to further adapt our model to a broader range of tasks.
>
> **Q4: The claim of SOTA performance on video-mme**
> **A4**: Thank you for your suggestion. We have revised all statements regarding SOTA in the manuscript, and now only highlight the significant improvements our method achieves over direct baselines employing fixed partitioning strategies. This adjustment helps readers better understand the focus and impact of our work.

---

### Official Review · Reviewer_LeaC · 2025-10-31

**Soundness:** 3
**Presentation:** 3
**Contribution:** 3
**Rating:** 4
**Confidence:** 3

**Summary:**

This paper introduces Quicksviewer, a method for efficient video understanding. The core idea is to dynamically and non-uniformly partition videos into variable-length cubes, enabling adaptive spatiotemporal token compression.

**Strengths:**

1. The use of a learnable cube network to dynamically and non-uniformly segment videos is both reasonable and novel.

2. The adoption of Gumbel-Softmax allows for unsupervised, end-to-end boundary learning, avoiding the need for manual annotation.

**Weaknesses:**

1. The claimed SoTA performance may be misleading, as the baseline methods used for comparison are relatively outdated given the rapid progress in long video understanding. While Table 2 reports Quicksviewer as SoTA, recent models on the Video-MME leaderboard (e.g., VideoChat-Flash, VideoLLaMA-3) achieve scores above 65 with comparable or even smaller model sizes, whereas Quicksviewer only achieves 56.9.

2. The dynamic partitioning is content-aware but not query-aware. For queries requiring fine-grained details, pre-partitioning may result in the loss of critical information that cannot be recovered.

3. More comprehensive comparisons with recent token compression and long video understanding methods are needed, along with deeper analysis.

**Questions:**

The main concern is the potentially misleading reporting of SoTA results. For a top-tier conference like ICLR, the authors should compare against the latest baselines, at least those released in the first half of 2025.

---

> ### Author Response · Authors · 2025-11-29
> **Rebuttal to Reviewer LeaC**
>
> Dear Reviewer LeaC, we appreciate your efforts in taking the time to review our paper and provide reviews. Below, we have provided explanations and improvements for our work according to your concerns:
>
>
> **Q1: The claim of SOTA performance on video-mme**
> **A1**: Thank you for your suggestion. We have revised all statements regarding SOTA in the manuscript, and now only highlight the significant improvements our method achieves over direct baselines employing fixed partitioning strategies. This adjustment helps readers better understand the focus and impact of our work.
>
> **Q2: The dynamic partitioning is content-aware but not query-aware**
> **A2**: Our partitioning strategy, which is based on the information density of the video itself, is query-agnostic—this is one of the key ideas of our work. As outlined in the Introduction, our motivation stems from two main considerations:
>   - (1) Most real-world video scenarios involve online video streams, where compression needs to be performed in advance before receiving user queries. Therefore, we propose a compression method based on the intrinsic information density of the video and calculate compression probabilities using temporal momentum, which eliminates redundant computations on historical frames and enables fully online compression.
>   - (2) During inference deployment, it is often necessary to pre-fill videos prior to user queries. Repartitioning videos for each query would incur additional overhead. The advantage of our approach is that it performs low-level, query-independent pre-compression, facilitating faster inference within deployment frameworks.
>
> **Q3: More comprehensive comparisons with recent works**
> **A3**:
>   - The primary research question of this work focuses on how to partition video cubes based on the intrinsic information density for dynamic compression, aiming to achieve efficient encoding while enhancing understanding through rational segmentation. Compared to a direct baseline that employs a fixed partitioning strategy, our model achieves an improvement of 8.7 percentage points, demonstrating the effectiveness of our approach.
> - We also conducted comparative experiments with VLMs of similar size. On Video-MME, our model achieves competitive performance with lower token consumption.
> - Furthermore, our model is a video model trained end-to-end from a pure language backbone, rather than leveraging a well-trained VLM backbone. In subsequent experiments, we implemented our Quicksviewer’s cubing module on a pre-trained MiniCPM-v 2.6 backbone and performed fine-tuning. The resulting model achieves **72.9 accuracy on Video-MME (no captions, max 420 frames)**, matching SOTA performance while significantly improving the base model.

---

### Official Review · Reviewer_iMvC · 2025-11-01

**Soundness:** 4
**Presentation:** 3
**Contribution:** 3
**Rating:** 4
**Confidence:** 5

**Summary:**

This paper introduces Quicksviewer, an efficient video understanding model designed to address the issues of low computational efficiency and temporal-spatial redundancy caused by uniform frame perception in large-scale multimodal models (LMMs). Quicksviewer proposes a new non-uniform perception paradigm: it first uses a "cubing network" and Gumbel Softmax to dynamically split the video into non-uniform "video blocks" based on the temporal density of the video content. Then, a unified resampler compresses each variable-length block into a fixed number of visual tokens. It achieves significant compression rates, allowing the model to efficiently process long videos at all training stages.

**Strengths:**

1. The model uses the Gumbel Softmax trick to integrate discrete "block networks" into end-to-end training.
2. The cubing that tracks semantic changes in momentum, making it applicable "online" to video stream scenarios.

**Weaknesses:**

1. The resampler forces the content of 5 frames and 50 frames into 64 tokens, which is a "distortion" with highly uneven information density. The paper also mentions "misalignment between the model's visual positional encoding and the actual temporal sequence," which undermines the model's ability to perform accurate temporal reasoning during inference. Is it possible to use dynamic token numbers, allowing information-rich blocks (e.g., 50-frame blocks) to be compressed into more tokens (e.g., 128), while information-poor blocks (e.g., 5-frame blocks) are compressed into fewer tokens (e.g., 32), thereby maintaining relatively constant information density.
2. The paper does not provide a direct comparison of speed with the original baseline (for example, processing a 10-minute video, Quicksviewer takes 30 seconds, while the baseline model takes 60 seconds), making it difficult for readers to intuitively understand whether Quicksviewer has additional computational overhead.
3. Methods may sacrifice some performance in fine-grained target understanding, which is also mentioned in the paper.
4. The paper focuses on video token compression but lacks comparison with other token compression works, such as visionzip, videoxl, etc.

**Questions:**

1. Have there been tests on temporal grounding tasks for long videos? Can this compression method be applied?
2. What is the maximum number of frames or length of video the model can support?

---

> ### Author Response · Authors · 2025-11-29
> **Rebuttal to Reviewer iMvC**
>
> Dear Reviewer iMvC, we appreciate your efforts in taking the time to review our paper and provide reviews. Below, we have provided explanations and improvements for our work according to your concerns:
>
> **Q1: The misalignment between visual PE and temporal sequence undermines model’s ability**
> **A1**:
>   - The major problem addressed in this paper focuses on how to divide videos with nonuniform information density into multiple cubes for efficient compression. We propose a solution by incorporating annealed Gumbel-softmax into the end-to-end training of large VLMs, and further empirically demonstrate its effectiveness.
>   - We have candidly discussed several limitations of our preliminary work in the LIMITATIONS section, such as the temporal misalignment between visual positional encoding (PE) and visual tokens. In this work, we have made initial attempts to address this issue: **(1) We appended timestamp strings (e.g., <cube_start>…<cube_end> at second 7 <cube_start>…) to the visual tokens of each cube, enabling the model to obtain an absolute sense of time for the visual tokens within each cube; (2) We adopted a fixed 5FPQ training protocol, which ensures that videos of the same duration contain approximately the same number of visual tokens, thereby helping the resampler learn a stable mapping under a fixed PE.**
>   - Moreover, we are actively working to improve the PE method in our ongoing research, and would like to share some of our preliminary solutions: (1) We first calculate the start and end time boundaries for each cube (e.g., 0–6 seconds for the first cube); (2) We then uniformly sample PE indices within the interval (e.g., np.linspace(0, 6, 64)); (3) These interpolated or extrapolated indices are finally used to compute the PE embeddings based on M-ROPE. This approach enables more precise temporal alignment, and our initial experiments have shown promising results. We have updated the relevant content in the LIMITATIONS section of the manuscript accordingly.
>
> **Q2: The comparison of speed with original baseline**
> **A2:**
>   - In accordance with your suggestion, we selected a 10-minute video from VideoMME (https://www.youtube.com/watch?v=Z7LJrU4443Q ) and tested our model alongside a video model of comparable size, LLaVA-Next-Video-Qwen2, under identical conditions. The inference environment for both models was kept consistent, with no quantization or inference acceleration frameworks used; only the HuggingFace inference framework was adopted. The results are presented in the following table:
>
> ----
> |Model | #Input Frames | First token response time |
> |--|--|--|
> | LLaVA-Next-Video-Qwen2  (8B) | 300  | 0.356 seconds |
> | Quicksviewer (8B) | 300  | 0.214 seconds |
> ----
>
> &nbsp;&nbsp; As shown in the table, our model achieves a significantly lower initial response time compared to the video model of the same size, owing to compression, while maintaining competitive performance.
>
> **Q3: The performance in fine-grained understanding**
> **A3:**
>   - The MiniCPM-4.5v paper has demonstrated that VLMs employing a 3D resampler can achieve higher frame rates (FPS) by adopting a greater compression ratio (from 64 to 21.3 tokens per frame), thereby enabling improved fine-grained motion understanding at a fixed sequence length (performance on VideoMME improves from 65.5 to 67.3). Building on the 3D resampler, our work further explores foundational compression at the video clip level, which raises the upper limit for VLM inference FPS and facilitates more effective modeling of fine-grained motion.
>   - Additionally, we fully acknowledge your concern regarding fine-grained small object recognition, and agree that this remains a fundamental challenge in video understanding. We believe it should be addressed by enhancing the VLM’s capacity to exploit spatial information within individual video frames.
>
> **Q4: Tests on temporal grounding tasks**
> **A4**: As our model is primarily designed to enhance video encoding efficiency, we did not evaluate its performance on the temporal grounding test set. However, the comprehensive Video-MME benchmark includes a “Temporal Perception” subtask, where our model, benefitting from the use of absolute timestamps, also demonstrates satisfactory performance.
>
> **Q5: the maximum number of frames Quicksviewer can support**
> **A5**: Without applying FP4/8 quantization, our model—using bfloat16—can process up to 960 frames on a single NVIDIA A100 80G GPU. With FP4 quantization, it can handle up to 3600 frames (i.e., 1-hour video at 1 FPS). With multi-GPU deployment, even longer videos can be processed.

---

### Meta-Review · Area_Chair_tLKx · 2026-01-06

**Summary:**

This paper proposes a framework for improving the efficiency of large multimodal models for long-video understanding by dynamically and non-uniformly compressing video tokens. The paper initially received overall rejection recommendations, specifically one "reject" (2) and two "marginally below the acceptance threshold" (4). Reviewers raised several substantive concerns, including the following:

1. Distortion and potential performance degradation induced by video token compression and misalignment between visual positional embeddings (PEs) and the temporal sequence.
2. Lack of direct comparison with recent token compression methods.
3. Insufficient evaluation of fine-grained target understanding.
4. Absence of a direct speed comparison with the original baseline.
5. Overstated claims that the proposed method achieves state-of-the-art (SOTA) performance.

**Reviewer Concerns:**

The authors addressed some concerns; however, the rebuttal was insufficient.

1. The authors acknowledged that the concern about the PE misalignment introduced by video token compression is valid and noted that it was originally discussed in the limitations section.
2. The authors did not provide a direct comparison with recent token compression methods. Instead, they presented additional results using more recent backbones.
3. The authors claimed that the proposed method demonstrates satisfactory performance on temporal grounding, a subtask of Video-MME. However, no objective experimental results were provided, and the discussion relied on vague qualitative statements.
4. In the response to Reviewer iMvC, the speed comparison was reported without accompanying performance metrics, with the authors merely stating that the proposed method maintains competitive performance.
5. The authors promised to tone down the SOTA claim in the revision.

Overall, the majority of the major concerns were not adequately addressed.

**Reviewer Scores:**

This paper presents a simple and potentially promising approach to efficient long-video understanding. However, the reviewers’ overall initial evaluations were negative, raising several substantive concerns. The authors promised to tone down their SOTA claims, but apart from this, most major concerns remain unresolved. In particular, the misalignment of positional embeddings (PEs) and the absence of comparisons with state-of-the-art token compression methods were identified as critical issues. Consequently, with this insufficient rebuttal, the reviewers would not change their initial ratings even if a full discussion period is given, and therefore, I recommend rejection.

---

### Decision · Program_Chairs · 2026-01-26

Reject